# Improving Pharmacokinetics of Peptides Using Phage Display

**DOI:** 10.3390/v16040570

**Published:** 2024-04-07

**Authors:** Mallika Asar, Jessica Newton-Northup, Mette Soendergaard

**Affiliations:** 1College of Osteopathic Medicine, Kansas City University, Kansas City, MO 64106, USA; mallika.asar@kansascity.edu; 2Cell Origins LLC, 1601 South Providence Road Columbia, Columbia, MO 65203, USA; jessica@cellorigins.com; 3Department of Chemistry, Western Illinois University, Macomb, IL 61455, USA

**Keywords:** phage display, peptides, pharmacokinetics, biopanning, affinity selection, nanoparticles

## Abstract

Phage display is a versatile method often used in the discovery of peptides that targets disease-related biomarkers. A major advantage of this technology is the ease and cost efficiency of affinity selection, also known as biopanning, to identify novel peptides. While it is relatively straightforward to identify peptides with optimal binding affinity, the pharmacokinetics of the selected peptides often prove to be suboptimal. Therefore, careful consideration of the experimental conditions, including the choice of using in vitro, in situ, or in vivo affinity selections, is essential in generating peptides with high affinity and specificity that also demonstrate desirable pharmacokinetics. Specifically, in vivo biopanning, or the combination of in vitro, in situ, and in vivo affinity selections, has been proven to influence the biodistribution and clearance of peptides and peptide-conjugated nanoparticles. Additionally, the marked difference in properties between peptides and nanoparticles must be considered. While peptide biodistribution depends primarily on physiochemical properties and can be modified by amino acid modifications, the size and shape of nanoparticles also affect both absorption and distribution. Thus, optimization of the desired pharmacokinetic properties should be an important consideration in biopanning strategies to enable the selection of peptides and peptide-conjugated nanoparticles that effectively target biomarkers in vivo.

## 1. Introduction

Phage display was pioneered by Dr. George Smith at the University of Missouri in 1985 [1]. The technology is a combinatorial technique that employs the use of an assembled library of filamentous bacteriophages, in which the DNA has been genetically modified, resulting in the fusion of a foreign peptide or antibody to the N-terminal end of one of the viral coat proteins [1,2,3,4,5,6,7]. Such phage libraries provide researchers with a vast pool from which polypeptide-based ligands that exhibit high affinity and specificity towards a target antigen can be selected. The versatility of phage display technology has made it invaluable in various applications, particularly in the discovery of therapeutic and diagnostic peptides. The technique has been used to develop ligands that exhibit high binding affinity and specificity for their target and that can be used in either their soluble form or conjugated to biological nanoparticles [6,8,9,10,11]. However, while equally important to ligand binding affinity and specificity, the pharmacokinetic properties are often overlooked when developing ligands using phage display technology. Here we argue that careful optimization of phage display affinity selections leads to the identification of peptides with desirable binding affinity, specificity, as well as pharmacokinetic properties for direct delivery to tissues. 

Biopanning of phage display libraries is the most widely used method for affinity selection and identification of peptides. In general terms, this procedure involves incubating a phage display library with the target antigen and subsequently eluting and collecting the bound phages. Although such affinity selections are a relatively straightforward and efficient technique, it necessitates the meticulous optimization of various parameters to obtain ligands with sufficient affinity and without off-target binding. These parameters encompass the selection of the phage display library type, as well as the concentrations of phage particles and antigen molecules. Additionally, factors such as antigen type (e.g., recombinant proteins, cell lines, whole tissues, organs, etc.), incubation conditions such as temperature and duration, and the method of phage amplification must be carefully considered [12]. 

Affinity selections are incredibly versatile as they allow for the screening of peptide libraries against a wide range of targets. In vitro biopanning experiments typically involve affinity selecting the phage display library against recombinant proteins or other isolated molecules, often leading to the identification of high-affinity binders. However, several research studies have shown that peptides selected through an in vitro phage display fail to exhibit optimal binding in live tissues and whole organisms, and their pharmacokinetic properties are often suboptimal [13]. Such ligands may bind to non-target tissues due to low specificity, resulting in possible side effects and reduced effectiveness when used for therapeutic purposes [14,15]. Furthermore, peptides often have low stability in vivo due to the presence of proteases that facilitate rapid degradation and thus limit the therapeutic potential [16,17]. For these reasons, it is important to design and optimize peptides to improve their stability and selectivity, which can be achieved through modifications such as cyclization and the incorporation of non-natural amino acids [18,19,20,21]. However, the selection of peptides with optimal pharmacokinetics may also be achieved by carefully designing the phage display biopanning experiments and using phage display for peptide maturation [22]. In particular, researchers are increasingly recognizing the advantages of in situ phage display against cells and tissues [23,24,25,26,27], as well as in vivo selections in live animals or patients [28,29,30,31]. Such biopanning strategies are better able to recapitulate the complex and treacherous milieu that diagnostic and therapeutic peptides are challenged with, and they should therefore be built into the phage display protocol as a selective pressure. The use of cell lines or animal models, or the combination of both, has proven particularly useful for developing and optimizing peptides that target biomarkers that are not easily replicated in vitro.

Enhancing the pharmacokinetics of peptides and peptide-conjugated biological nanoparticles can be achieved by utilizing phage display to select peptide sequences that exhibit the following: (1) enhanced binding affinity, (2) target specificity, and (3) desired biodistribution and clearance. Strategies to optimize binding affinity have been intensely discussed in other studies [12,32,33,34]. Here, we discuss how phage display technology can be used to enhance target specificity, and the biodistribution and clearance of peptides that are used in the targeting of disease-relevant biomarkers. Specifically, the utilization of biopanning strategies that recapitulate the complex milieu in vivo leads to the identification of peptides with such desired properties. 

## 2. Filamentous Bacteriophages

Ff-specific phages are a subgroup of the filamentous phage family that infects Gram-negative bacteria, such as *Escherichia coli*, carrying the F plasmid. These bacteriophages belong to the *Inoviridae* family and *Inovirus* genus, and they include the M13 and fd species. Phage display technology harnesses the unique characteristics and replication cycle of these filamentous bacteriophages. Specifically, fd and M13 are widely employed in phage display applications [35]. The Ff group of filamentous bacteriophages possesses a ssDNA genome comprising 98% similarity among the different strains. Both the fd and M13 species contain a 6.4 kb ssDNA genome that comprises 11 genes, and the virions measure approximately 6.5 nm in diameter and 930 nm in length. The genes are categorized into groups based on the functions of the corresponding proteins, including the following: (i) capsid proteins (pIII, pVI, pVII, pVIII, and pIX), (ii) DNA replication proteins (pII, pV, and pX), and (iii) proteins involved in virion assembly (pI, pIV, and pXI). The minor coat proteins, pIII and pVI, are found at one end of the rod-shaped phage virion, while the other end displays the minor coat proteins, pVII, and pIX [36,37]. Thousands of copies of the major capsid protein, pVIII, form the body of the filamentous phage capsid surrounding the ssDNA genome [38].

The filamentous bacteriophage infection of *E. coli* induces a lysogenic state wherein infected bacteria assemble and release progeny into the growth medium under cultured laboratory conditions. Infection occurs through the attachment of the phage minor coat protein pIII to the F pilus of a male *E. coli* cell. During this phase, the circular ssDNA is transferred to the bacterial cell, where it is transformed into a double-stranded plasmid by the replication machinery of the host cell. Rolling circle replication produces ssDNA that encodes the proteins required for packaging the DNA into the viral capsid. The fully assembled bacteriophages exit the bacterium without lysing the cell, which is a tremendous advantage of filamentous phages regarding phage display technology [35,39].

## 3. Principles of Phage Display

The first phage display system was developed using the fd–tet filamentous phage by George Smith in 1985 [1]. While numerous phage display systems have been constructed since then, those based on the Ff phages, fd and M13, remain the most prevalent and widely utilized [35]. This can be attributed in part to the simplicity of the phage structure with minor coat proteins on each end of the filamentous virion and the major coat protein along the body of the capsid. In addition, the phage genome is simple to manipulate, and the phage particles are easily propagated in bacterial cells, enabling the generation of a vast number of virions in a short period [39]. The latter is of particular importance in comparison to lytic phages that exhibit more complicated propagation in culture due to bacterial cell killing.

A key component of phage display technology involves the insertion of a foreign sequence at a specific position within a functional viral gene. Importantly, the insertion must maintain the functionality of the protein product and lead to the display of the expressed foreign sequence as a fusion protein on the viral surface. A phage display library is constructed by inserting foreign sequences of random nucleotides, creating a collection of potentially over 10^10^ distinct peptides, each displayed as a coat protein fusion with the phage particle [40]. The extensive variability enables the construction of diverse libraries of peptides, proteins, antibody fragments, and enzymes.

### 3.1. Phage Display Systems

The filamentous phage structure provides numerous possibilities for displaying polypeptide-based ligands, including peptides, antibodies, and antibody fragments (Table 1). While the most common approach is to express foreign ligands on coat proteins pIII, each coat protein utilization offers unique advantages. The pIII protein is located at the tip of the phage and is typically expressed in five copies [41]. The N-terminal domain of pIII initiates the transduction of viral DNA into *E. coli* during infection and is responsible for binding to the F pilus of male *E. coli* [41]. For phage display technology, the most commonly used expression system displays antibodies and peptides on the N-terminus of pIII and is typically separated from the wild-type (WT) protein by a short peptide linker [42,43]. A phage display system with pIII results in the expression of 3–5 copies of the foreign peptide at the phage tip when the bacteriophage vectors that have been built on the fd or M13 phage genomes are used [44]. Phagemid vectors that are smaller and lack most of the WT phage genes have been created that allow for the display of a single peptide on the N-terminus of pIII. Remarkably, the N-terminus of the pIII protein can accommodate relatively large molecules, up to 38 amino acids, without affecting phage infectivity [45].

Second in popularity to the pIII phage display is display on the major coat protein VIII (pVIII). Around 2700 copies of the pVIII protein are tightly arranged along the phage surface [35]. While not all the copies of pVIII are typically utilized for phage display, the expression of a substantial number (200–300) of peptide copies can be achieved by a type 88 system that expresses both the WT pVIII gene and the recombinant fusion peptide-pVIII protein [46,47]. This type of phage display offers distinct advantages in certain selection strategies, especially those targeting polyvalent antigens. Nonetheless, the avidity effect of expressing in the hundreds of peptides can lead to selecting peptides with lower binding affinity compared to ligands that were selected using the oligo- or monovalent display. The pVIII display of large polypeptides, such as full-length monoclonal antibodies and antibody fragments, often presents challenges as it can severely impede the release of virions from the *E. coli* host cell during the phage lifecycle [48,49]. However, shorter peptides can be successfully expressed. In addition to the pIII and pVIII display systems, the other filamentous phage coat proteins have also been employed in phage display to present foreign peptides and small proteins. Coat protein VI (pVI) is located at the tip of the phage virion adjacent to pIII. While far less common than the pIII display, pVI has proven advantageous in expressing larger proteins at both the N- and C-termini. Notably, the utilization of pVI expression in creating cDNA libraries provides distinct advantages compared to similar methods such as yeast-2-hybrid [50]. Coat protein VII (pVII) is positioned at the opposite end of the phage virion relative to pIII and pVI. The utilization of pVII for displaying antibodies and antibody fragments has been receiving increasing attention in recent years. Furthermore, combining the pVII display with the other types of expression systems (i.e., pIII) can facilitate tagging and post-screening analysis. Recent improvements have significantly enhanced the display of the coat protein IX (pIX) by leveraging diverse phagemid systems and signal sequences. These pIX systems tend to exhibit lower expression levels, which augments the probability of monovalent display and thus enables the selection of high-affinity ligands [9,51,52,53].

### 3.2. Bacteriophage Vectors and Phagemids

The original phage display libraries created by Dr. George Smith utilized the genetic modification of the fd filamentous phage genome to incorporate foreign peptide sequences onto pIII [1,54,55]. These and the M13 bacteriophage vectors would encode a genetically engineered version of the complete phage genome. Along with the wild-type genes, most bacteriophage vectors also contain an antibiotic selection gene, as well as a recombinant fusion gene that encodes the foreign, displayed peptide or antibody fused to a coat protein. Such type 3 + 3 systems can be used for displaying peptides, antibodies, or antibody fragments with minimal impairment of the *E. coli* infection [54,55,56]. An advantage of bacteriophage vectors is that they do not require the use of helper phage for viral propagation, as the genome contains all the necessary replication genes. However, the large size of antibody sequences can hinder proper transformation of the phage genome into *E. coli* by the heat-shock method and electroporation. For this reason, the traditional bacteriophage vectors are most often used for peptide display. 

Since the initial development of phage display, a second type of vector has gained significant popularity. Phagemids are plasmid-based vectors that combine phage and bacterial replication origins, along with an antibiotic selection gene [57]. These versatile vectors enable the display of full-length antibodies, antibody fragments, and peptides due to the small size of the DNA molecule (<4000 bp) and the common practice of propagating virions from the phagemid DNA rather than relying on an *E. coli* infection. However, phagemids lack most of the genes necessary for virion assembly and release, and for this reason, a helper phage that carries these genes is required for the production of viral progeny. Furthermore, the compact size of phagemid vectors simplifies their genetic manipulation and makes it possible to insert large sequences, and they have, for this reason, become the vector of choice for the display of large proteins such as full-length antibodies and antibody fragments [58].

### 3.3. Affinity Selection (Biopanning)

Affinity selection, also known as biopanning, of phage libraries is a widely employed technique for selecting polypeptide-based ligands that exhibit specific binding to a target antigen. The traditional biopanning procedure involves incubating a phage display library with the desired target, followed by the elution of bound phages. Each biopanning protocol comprises crucial steps that must be carefully considered [12].

The initial stage of biopanning is the actual step of affinity selection, wherein the phage display library is incubated with the target molecule. Phage clones with affinity for the target are allowed to bind by ensuring that the experimental conditions sufficiently facilitate the interaction [59]. This can be achieved by adjusting the temperature, duration, pH, ionic strength, detergent concentrations, etc. of the selection. Most biopanning protocols are performed in vitro against immobilized recombinant proteins; however, it is also possible to use carbohydrates and other biological and non-biological molecules as targets [60,61,62]. Moreover, successful biopanning protocols have made use of in situ and in vivo strategies that target whole cells, tissues, and organisms (Figure 1) [22,31,63,64,65,66].

When performing affinity selections, it is crucial to consider the level of stringency in the selection process. This can be optimized by modifying the experimental conditions as described above and by optimizing the concentration of either the phage library or the target antigens [12]. For instance, to isolate high-affinity ligands, one can enhance the competition between different phage clones by increasing the concentration of the phage library or decreasing the concentration of the target molecule. By implementing such rigorous conditions, the probability of identifying desirable phage clones is significantly enhanced. Biopanning procedures commonly involve multiple rounds of affinity selection. Typically, four rounds of biopanning are used to ensure a substantial selection pressure. Moreover, biopanning often comprises various types of affinity selections to identify phage clones with specific binding characteristics. One such approach involves combining the phage display selection against a recombinant protein with a selection round using a cell line that expresses the target protein on the cell surface, or vice versa. Such multi-tier biopanning strategies are used to increase the probability of selecting ligands that exhibit optimal binding properties in both in vitro and in vivo settings [22].

After the affinity selection step, unbound phages are usually removed by washing with a buffer containing a low concentration of detergent. Weakly bound phages, which are generally undesirable in phage display selection, can be eliminated by either increasing the detergent concentration or adding extra washing steps. In standard biopanning procedures, the collection of bound phages is accomplished through elution using detergents, pH changes, or other methods that disrupt the non-covalent interactions between the phage and target. These elution methods generally recover the majority of the bound phages. However, phage clones with high binding affinity may not be effectively eluted through the disruption of non-covalent interactions alone. Therefore, phage display libraries that contain a trypsin digestion site between the displayed peptide or antibody and the coat protein have become widely used. This feature facilitates trypsin elution that is indiscriminate of the binding affinity between the displayed ligand and the target molecule [12,67,68]. 

After each selection round, eluted phages are usually amplified and used in subsequent selections. Amplification of the collected phages can be conducted in different ways and depends on the type of vector used for the phage library. Bacteriophage vectors, such as the fUSE5 and f3TR1 libraries, are most often amplified by the infection of *E. coli*. Phagemid particles can also infect *E. coli*; however, require a helper phage for the propagation of viral particles. In addition, phagemid vectors are also easily transformed into *E. coli* as plasmids for amplification [12]. Amplification of the affinity-selected phage in *E. coli* is an easy and efficient method. However, the infection of *E. coli* can be influenced by the phage-displayed peptide or antibody. As described above, the infection of *E. coli* is facilitated by the interaction between pIII and the bacterial F pili. The utilization of the widely used pIII display libraries can potentially alter the infection efficiency of *E. coli* and introduce a selection bias towards specific phage clones [69,70]. Similar selection biases can also arise with the other display systems (pVI, pVII, pVIII, and pIX), where the displayed peptides or antibodies may impact various stages of the filamentous phage lifecycle, including the assembly and release of progeny. These biases can be overcome by employing PCR to amplify the sequences that encode the foreign peptide or antibody. Subsequently, the PCR amplicons can be reintroduced into the bacteriophage vector or phagemid for transformation and propagation in *E. coli*.

The phage clones collected from the affinity selections are identified through DNA sequencing of the foreign inserts that encode the phage-displayed peptides or antibodies. Typically, only the phage clones from the final round of affinity selection undergo DNA sequencing. However, there can be advantages to sequencing the phage mixtures after each round of selection. This approach allows the researcher to track the efficiency of the phage display procedure, as phage particles with binding affinity for the target should become more abundant as the selection proceeds. Traditionally, phages have been identified by Sanger sequencing of individual *E. coli* colonies infected by the selected clones. However, this method is tedious and low throughput. Instead, most researchers now take advantage of next-generation sequencing (NGS) to identify millions of phage clones. Furthermore, the large amount of data that are obtained in this manner can be used to sort identified sequences based on motifs as well as other characteristics [71,72,73].

### 3.4. Peptide Phage Display

Historically, peptides were selected for high affinity and specificity within an in vitro environment and only then tested within an in vivo environment. This methodology has produced mixed results [74,75,76]. Thus, for the past ~20 years in vivo phage display selections have been more widely utilized to select for peptides that optimally target tissues and organs within the milieu of a live animal [77,78,79,80,81]. More recently, in vivo phage selections have been performed to identify peptides with the capability of modifying biodistribution and clearance of themselves and/or larger particles [78,82,83,84]. These phage display-selected peptides are often discovered to have specific capabilities such as extravasation out of the vasculature and into specific tissues, extended serum/blood half-lives, or preference clearance by a specific organ [77,82,84]. Each type of biopanning strategy encompasses its own advantages and disadvantages, which are summarized in Table 2 and further discussed in the next sections. Overall, in vitro selections against recombinant proteins and other isolated biomolecules result in peptides with relatively low K_d_-values compared to the other methods, but they also suffer from poor pharmacokinetics and stability. The opposite is true when employing in vivo selections in live animals or patients. This type of biopanning typically leads to enhanced pharmacokinetics, stability, and specificity, but it also often produces higher K_d_-values compared to in vitro strategies. However, the binding kinetics can be substantially improved by affinity maturation after the initial phage display selections. In situ biopanning provides advantages from each of the aforementioned strategies, including low-to-mid range K_d_, enhanced binding specificity, and stability. However, this type of method can fail to discover peptides with optimal pharmacokinetics. Often, it is an advantage to combine two or all three of these biopanning methods in a so-called multi-tier strategy to develop peptides that exhibit all the desired properties such as high binding affinity and specificity, as well as enhanced stability and pharmacokinetics.

#### 3.4.1. In Vitro Selected Peptides

Selection of peptides within the in vitro environment often aids in the discovery and development of high-affinity and high-specificity peptides. Examples of this are numerous, including a recent study by Díaz-Perlas et al. [85] that utilized modified versions of three fd–tet vector-based phage display libraries to select peptide ligands against isolated calprotectin. The libraries were subjected to parallel affinity selections revealing sequence similarities between the top identified peptides. One peptide was chosen for X-ray crystallography and alanine scanning experiments that showed that the peptide engaged with calprotectin via hydrogen bonding and the interaction of hydrophobic residues with cavities on the surface of the target protein. Binding analyses revealed a high binding affinity in the lower nanomolar range (K_d_ = 26 ± 3 nM) emphasizing the utility of carefully designed phage display selections [85]. Nevertheless, phage display selections against over-simplified in vitro targets, such as recombinant proteins, are often unable to predict pharmacokinetic properties in vivo. An example of an in vitro selected peptide with proven utility in vivo is the anti-galectin-3 peptide, G3-C12. This phage display-selected peptide was originally found and characterized by Dr. S. L. Deutscher’s laboratory [86,87]. It has since been utilized by at least two groups to molecularly target N-(2-hydroxypropyl) methacrylamide (HPMA) copolymers to various types of galectin-3 overexpressing tumors, including colon and prostate carcinomas [88,89,90,91]. The G3-C12 peptide is intriguing due to its multi-functional target and the fact that the peptide itself seems to have therapeutic potential. The target, galectin-3, is multifunctional and has been shown to switch between pro-apoptotic and anti-apoptotic properties, as well as possessing intricate roles in cell–cell adhesion necessary for metastasis [86,90]. Furthermore, the G3-C12 peptide has been shown to inhibit cell–cell adhesion via the galectin-3-carbohydrate binding function and reduce galectin-3 multimerization [86,87]. Additionally, Dr. Y. Huang’s group has detailed the use of G3-C12 peptide covalently attached to the HPMA copolymers loaded with doxorubicin and its ability to bind extracellular galectin-3, be internalized, and localized within the mitochondria forcing the balance of biochemical pathways towards apoptosis [90]. While peptide G3-C12 is successful in its own right, the high abundance cell surface target protein, gal-3, is expressed on the surface of endothelial cells and cancerous cells within tumors with leaky vasculature. Unfortunately, Gal-3 is also highly expressed on the surface of the tubules within the kidney, leading to higher-than-normal kidney uptake of the peptide [92,93,94]. Thus, the increased kidney uptake complicates the in vivo utility of the G3-C12 peptide as a gal-3 targeting agent.

Largely, in vitro phage display is apt at selecting peptides with K_d_-values in the lower nanomolar range and that can be used in the targeting of relevant disease biomarkers. However, this type of biopanning fails to incorporate tissue specificity and the treacherous nature of the in vivo environment that a diagnostic or therapeutic peptide is challenged with. 

#### 3.4.2. In Situ Selected Peptides

In comparison, in situ phage display selections against cells or tissues are viewed by many as advantageous for the ability to better recapitulate the complex in vivo milieu and to discover new cancer-specific targets. An increasingly popular choice in the design and implementation of such phage display selections is the use of subtractive selections [82,95,96]. One such study was conducted by Asar et al., who utilized a subtractive in situ phage display strategy to select for peptides against human pancreatic cancer cells. In this investigation, a 15-mer fUSE5 library was first negatively selected by incubation with normal immortalized pancreatic cells (hTERT-HPNE), followed by four positive rounds of increasing stringency against the pancreatic ductal adenocarcinoma cell line, Mia Paca-2. Next-generation DNA sequencing of the negative and positive selections, combined with bioinformatic analysis, identified the peptide MCA1. The specificity of the peptide for the pancreatic cancer cells was validated by modified ELISA and fluorescent microscopy experiments using the hTERT-HPNE, embryonic kidney (HEK 293), ovarian (SKOV-3), and prostate cancer (LNCaP) cell lines [97]. This specificity was later conferred showing that the MCA1 peptide demonstrates no binding to the pancreatic cancer cell lines, Panc 10.05, CFPAC-1, and HPAF-II, thus exhibiting binding specificity to the selector Mia Paca-2 cells [98]. These studies emphasize the utilization of subtractive phage display selections combined with deep next-generation DNA sequencing when developing peptides with a high degree of specificity.

In situ phage display has also been successfully used to select for peptides that can target cancer cells in xenografted mouse models. Phage display has frequently been employed to create peptide-based targeting molecules with a high affinity for various types of cancer. However, validation of the selected peptides is often time-consuming and costly when translating these targeting agents for in vivo use. In contrast, peptide-displaying phages can be rapidly and affordably analyzed. Thus, phage display selections followed by binding analyses of the identified phage clones rather than the soluble peptides can save time and costs. Soendergaard et al. used a subtractive phage display affinity selection to identify phage clones with specific binding to the ovarian cancer cell line, SKOV-3. A fUSE5 15-mer peptide library was first pre-cleared against normal ovarian cells (HS-832), and then subjected to positive selections by incubation with ovarian adenocarcinoma (SKOV-3) cells. Micropanning, cell binding, and fluorescence microscopy assays identified two phage clones (M6 and M9) with high binding affinity and specificity to SKOV-3 cells. The SKOV-3 targeting of the phage clones was further validated in vivo using a xenografted mouse model. Both fluorescently labeled phages demonstrated tumor imaging capabilities in vivo with desirable biodistribution [99]. 

Tumor-associated M2 macrophages are a crucial part of the tumor microenvironment and thus an interesting target in the development of immunotherapies. Recently, Sioud and Zhang used an in situ phage display selection strategy to identify peptides that bind to M2 macrophages to explore the therapeutic potential of peptide–photosensitizer conjugates. Initially, subtractive affinity selection was carried out by first negatively selecting the library against healthy donor peripheral blood mononuclear cells (PBMCs) and M1 macrophages, followed by three rounds of positive selections against isolated M2 macrophages. However, this strategy resulted in phage clones that bound to prohibitin, which is expressed in both M1 and M2 cells. To facilitate the isolation of M2-specific peptides, the group instead blocked the cells with a known prohibitin-binding peptide, which resulted in the selection of new peptides that bound specifically to the M2 macrophages. The selected KML peptide conjugated with an IR700 photosensitizer was used to target M2 cells to induce photocytotoxicity in the cultured cells [27]. 

Careful consideration of the experimental parameters of in situ selections can lead to the discovery of peptides with enhanced binding specificity, as evidenced by the subtractive phage display selections using target and non-target cell lines. Nonetheless, the specific conditions of the negative and positive selection rounds must be optimized to avoid the carry-over of phage clones, leading to the peptides that bind a biomarker present in both the target and non-target cells. Most often, this can be avoided by optimizing the incubation times and concentration of virions in each round of biopanning. 

#### 3.4.3. In Vivo Selected Peptides

The challenges facing a therapeutic or diagnostic peptide from the time of injection to reaching and binding to the target are many. Of priority are maintaining sufficient stability, exhibiting only minimal off-target binding, and extravasation into the intended tissue. While in situ phage display selections better mimic the in vivo environment compared to in vitro biopanning strategies using isolated molecules, only selections in live animals or patients can truly recapitulate these challenges.

Two examples of novel targeting peptides from an in vivo selection protocol are G1 and H5 peptides, both specific to prostate adenocarcinoma [65,82]. Both peptides possess interesting abilities to induce unique cellular responses upon binding to the cell surface. Both of the peptides/phage clones were selected in vivo within PC-3 human prostate tumor-bearing mice using a phage library depleted from the vasculature binding phage. The selected phage clones were then further screened by parallel micropanning experiments to identify specific phage clones with the highest tumor-to-normal tissue binding ratios. The phage clones were then fluorescently labeled with AlexaFluor680, a near-infrared fluorophore, and the biodistribution of the labeled phage was investigated via optical imaging of live mice and ex vivo biodistribution to verify tumor specificity [82]. Microscopic investigation of fluorescently labeled phage clones and peptides revealed that these in vivo-selected phage clones were able to extravasate the vasculature, bind directly to the PC-3 tumor tissue, be internalized by the PC-3 cells, and activate apoptosis via caspase [65,82].

Another example of an in vivo-selected peptide is the RCC1-02 peptide [78]. This peptide resulted from a selection designed to redirect the biodistribution of the 26 MDa phage virion towards clearance through the kidney, instead of the liver and other organs of the reticuloendothelial system [100]. In short, a phage library was intravenously injected, and phages were collected from the urine of the mice. The stringency of the protocol was very high, and the resulting recovery was very low (10.6% to 10.9% for two different protocols). The lead clone, RCC1-02, was found to increase kidney uptake of the phage clone by 2.46-fold when compared to a phage clone with no foreign peptide displayed. Importantly, in vitro studies showed that the RCC1-02 phage clone as well as the RCC1-02 peptide were able to avoid reabsorption by OK proximal tubule cells.

In comparison to other biopanning methods, in vivo phage display selections reiterate the challenges facing diagnostic and therapeutic peptides regarding optimal pharmacokinetics and stability. Even so, the experimental parameters must be taken into careful consideration to avoid off-target binding. Importantly, the phage display library should be pre-cleared against non-target tissues, such as in a normal mouse, to enhance target specificity. The pharmacokinetic properties can then be selected for during positive selection rounds in a mouse model of the relevant disease by varying the incubation time and the concentration of injected virions. 

#### 3.4.4. Peptide-Conjugated Nanoparticles

The use and utility of phage display-selected peptides have also been expanded into the directed targeting of larger, more complex biomaterials and other moieties, such as nanoparticles (including liposomes and exosomes), peptide–protein conjugates, and polymers. An initial search of Google Scholar for “nanoparticles phage display peptide” resulted in 26,900 hits. The types of peptides conjugated to and incorporated in nanoparticles can be roughly divided into the following two categories: (1) peptides modifying the pharmacokinetics, such as the absorbance or clearance, of the nanoparticle versus (2) the pharmacodynamics of the nanoparticle such as targeting peptides [34,101,102]. Other types of nanoparticles include tin oxide, iron oxide, titanium oxide, gold, and other core chemistries [60,103,104,105,106,107,108]. Still more nanoparticles incorporate aqueous silica nanoparticles, virus-like particles, and the list goes on [109,110,111,112]. Almost all of these have been utilized with a phage display-derived peptide. Here, we present and highlight a small number of studies to illustrate the opportunities and potential uses of phage display-selected peptides for the molecular targeting of larger moieties. 

Mammalian cells produce and secrete extracellular vesicles that contain a mix of proteins and nucleic acids. These vesicles are believed to play an intricate role in cell-to-cell communications [113,114,115]. The idea of using lipid bilayer membrane vesicles for the delivery of bioactive molecules to cells has been readily adopted by the drug delivery field of research [116,117]. The following two different methodologies are currently utilized: that of liposomes, and that of exosomes [118,119]. Liposomes and exosomes are structurally similar, both are composed of a lipid bilayer. However, liposomes contain a limited number of lipids and no cellular protein or genetic material. In comparison, exosomes are more complex, with a wider variety of lipid bilayer components and cellular bioactive materials (proteins, nucleic acids, etc.). To date, liposomes and exosomes are both most often produced at ~100 nm in diameter. Both can have hydrophilic drugs packaged within the lumen of the vesicle and both can have hydrophobic drugs packaged in the lipid bilayer of the vesicle. Pertinent to this review is the use of targeting peptides and ligands on both types of vesicles. The potential delivery of various forms of RNA, peptides, and synthetic drugs via liposomes or exosomes has, however, been hampered by rapid off-target accumulation within the liver, kidney, and spleen [120]. Thus, the improvement of these liposomes and exosomes has been an attractive field for the use of phage display-selected targeting peptides. To this point, it is important to note that the structures, organization, and presentation of a peptide are strongly influenced by the microenvironment created by the fusion of the peptide to the coat protein upon the surface of a phage virion. This in turn impacts the binding of the free synthetic peptide versus a phage-displayed peptide. Thus, a soluble peptide with the same amino acid sequence may have very different binding characteristics than that of the same sequence fused to a coat protein. Added drawbacks to the use of synthesized peptides for imaging and therapy are the covalent conjugation of drugs, chelators, and fluorescent tags. Each of these can reduce the binding affinity of a peptide for its target [82,83]. Consequently, some researchers have formulated novel techniques to maintain the affinity of a phage display-selected peptide. These advances further the utility of an even greater variety of phage display-derived peptides.

Dr. K.C. Brown and their team have developed a tetrameric display of peptides upon a trilysine dendrimeric core [121,122]. They have shown that maintaining the valency and orientation of peptides from the original vector (in this case, the pIII display) within the tetrameric peptide, in a similar fashion as the surface of a phage particle, preserves the affinity of the phage display-selected peptides. Furthermore, they have successfully employed this technique for the molecular targeting of liposomes [122]. They compared K_D_ values of liposomes loaded with similar numbers of peptides but in a monomeric versus tetrameric presentation. Their data revealed a significant improvement in the K_D_ value from 9.2 nM for liposomes with monomerically displayed peptides versus 11 pM for peptides presented in a tetrameric format with a similar total number of peptides. Thus, they were able to deliver doxorubicin-loaded liposomes more efficiently leading to six times more toxicity.

In comparison, a collaboration between Drs. V.A. Petrenko and V.P. Torchilin have resulted in the design and use of the entire cp8 phage protein with the fused foreign peptide for the targeted delivery of multiple drugs, including doxorubicin [123,124,125,126]. These researchers found that the inherent physical and chemical properties of the filamentous phage coat protein VIII lend themselves well to use as a membrane protein within a liposome. Thus, they utilized landscape phage display (phage with all ~3000 copies of coat protein VIII genetically modified with the display of a foreign peptide sequence) for the selection of tumor-targeting peptides. In this way, they were able to avoid unexpected and undesirable side effects of chemical conjugation chemistries upon peptides [127]. Furthermore, they were able to show that the pVIII phage protein behaves in a fashion similar to known fusogenic peptides that facilitate the endosomal escape via structural changes due to acidification. This endosomal escape of the drug-loaded liposome results in an enhanced cytotoxicity to the targeted tumor cells.

An advantage of using phage display to identify peptides that are to be used conjugated to nanoparticles is the nanoscale size of the phage virion. The large size mimics the size and to some extent, the shape of many nanoparticles and thus the pharmacokinetics of these. To this point, a landscape phage can be used to match the physiochemical properties of the intended nanoparticle by changing the pVIII-displayed peptide to tune the overall charge and hydrophobicity of the phage particle. 

## 4. Pharmacokinetic Principles

### 4.1. Pharmacokinetics of Peptides

Pharmacokinetic studies focus primarily on distribution, absorption, metabolism, and elimination with a minor focus on drug-specific effects upon these attributes. For example, target-mediated drug disposition and immunogenicity. All these things can also be impacted by the route of administration.

#### 4.1.1. Routes of Administration

Peptides have gained increasing attention as a promising class of therapeutic drugs due to their specificity and potency for targeting disease pathways [128]. However, their short half-life in the bloodstream and potential for pre-systemic degradation pose challenges for effective delivery. Further, different routes of administration each present its unique challenges [129]. Parenteral administration, although common, has its limitations for peptide drugs. This route requires repeated dosing to maintain therapeutic levels, and there is potential for protease or peptidase activity to degrade the drug before it reaches its target. The development of effective oral peptide drugs has been an ongoing challenge due to the complexity of formulating a compound that can survive the harsh environment of the gastrointestinal tract and be absorbed into the bloodstream. Desmopressin and cyclosporine are currently the only two marketed oral peptide drugs, with their unique chemical properties contributing to their relatively high bioavailability [128]. Buccal delivery offers a promising alternative to oral delivery by providing direct absorption into the circulation, while bypassing hepatic metabolism and degradation in the gastrointestinal tract. However, taste and convenience remain significant drawbacks, as the dosing process can be cumbersome and unappealing to patients. 

#### 4.1.2. Distribution and Absorption

The manner in which drugs are distributed depends on their physiochemical and transport properties. Small peptide molecules mostly rely on passive distribution, while larger peptide molecules require convective transport [128]. The volume of distribution of peptides is restricted to the extracellular space. When administered intravenously, most peptides display a biexponential plasma concentration–time profile [130]. One factor that can affect peptide distribution is their binding to endogenous proteins. It is important to consider these aspects when designing drug delivery systems since understanding the mode of distribution can help increase drug efficacy and minimize potential side effects. To increase the absorption rates of non-invasive delivery routes, various strategies have been employed, including modifications to the amino acid backbone, formulation approaches, chemical conjugation with hydrophobic or targeting ligands, and the use of permeation enhancers [131]. 

#### 4.1.3. Clearance

For therapeutic peptides to be eliminated from the body, they either enter the metabolic pathways or undergo renal or biliary excretion [128]. Renal elimination is the predominate route of peptide clearance. Depending upon the size of the peptide, the rate of clearance can almost match that of the glomerular filtration rate [132]. However, renal clearance may be limited due to the glomerular filtration rate or proteolytic degradation in the kidney tubules. Some peptide drugs may also undergo hepatic metabolism for elimination, although intracellular uptake may pose limitations. For predominantly hydrophobic peptides a combination of passive diffusion and carrier-mediated uptake is the major uptake mechanism within the liver [133]. An example of hepatic clearance due to passive diffusion is that of cyclosporine. In comparison, octreotide uptake is facilitated by carrier-mediated transport [134]. The impact of protease and peptidase activities on peptide pharmacokinetics is hard to overstate. Different tissues contain varying levels of these proteolytic enzymes. Specifically, peptide stability should be examined within the blood, liver, kidneys, and small intestine because these tissues contain a large number of various proteases and peptidases. Most commonly, amino acid substitution and modifications are used to confer resistance to peptidases and proteases. 

#### 4.1.4. Drug-Specific Issues

A unique pharmacokinetic characteristic known as target-mediated drug disposition can affect how biologics, including peptides, are distributed and eliminated from the body [135]. Target-mediated drug disposition can produce a nonlinear pharmacokinetic profile. For example, some antibodies are known to have a significant proportion of the drug dose bound with high affinity to the pharmacological target, which in turn affects the subsequent elimination profile of the drug [136]. Three examples of peptides with target-mediated drug disposition include thrombopoietin mimetic peptide (PEG-TPOm), thrombopoietin mimetic peptibody (romiplostim), and peptidic erythropoiesis receptor agonist (ERA) [137,138,139]. Therefore, understanding the various routes of elimination for therapeutic peptides and the impact of target-mediated drug disposition is crucial in the development and use of peptide drugs.

Immunogenicity pertains to the undesirable immune response that may be caused by a therapeutic agent. Anti-drug antibodies (ADA) may arise upon recurrent or extended administration, potentially altering the pharmacokinetics of the drug and triggering hypersensitivity or anaphylaxis [140]. Though small peptides are typically inadequate immunogens, certain exceptions occur due to human immunity complexity. Immunogenicity is influenced by the method of administration, with subcutaneous injection being more likely to trigger an immune response than other routes [141,142]. Strategies to diminish peptide immunogenicity include avoiding antigenic sequences in amino acid composition and incorporating modifications, such as glycosylation and PEGylation.

### 4.2. Pharmacokinetics of Phage and Nanoparticles 

Pharmacokinetic studies of larger particles like filamentous phage or other nanoparticles can be more complex due to the large size and molecular complexity of the particles. However, these studies still focus primarily on administration, distribution, metabolism, and elimination. Furthermore, the size and shape of each type of nanoparticle and phage strongly impact the absorption and biodistribution within the vasculature and into other body tissues. The metabolism and excretion of a large particle are generally viewed in the context of “clearance”, which might be better defined as the inactivation of a nanoparticle or phage via metabolism and the subsequent removal via excretion.

#### 4.2.1. Pharmacokinetic Profile of Filamentous Phage 

Phages have been administered to mice and other animals via numerous routes including orally, intranasally, intravenously, intraperitoneally, topically, and more (Table 1). However, here, we focus on the fate of intravenously injected filamentous fd or M13 phage. The biodistribution of filamentous phage, specifically the fd and M13 phage, includes an extended blood half-life and clearance by the reticuloendothelial system [100,143,144]. 

#### 4.2.2. Distribution and Absorption 

Multiple articles report a blood half-life of the phage of around 30 min with tissue accumulation starting as early as about 5 min [100,144]. However, the clarity of this is complicated by the continuous but low levels of phage in the blood even at 24 h. The blood-to-tissue ratios of various timepoints throughout the first 24 h post-injection strongly suggest that extravasated phages are not necessarily retained within the tissues [100,144]. Also, the presence of a displayed foreign peptide influences the overall biodistribution; however, the accumulation, retention, and clearance of phage by the organs of the reticuloendothelial system are difficult to overcome [78]. Further, phages can extravasate the vasculature and penetrate the surrounding tissues [100]. Molenaar et al. reported predominate uptake of ^35^S-methionine and ^35^S-cysteine radiolabeled M13 phage by the reticuloendothelial system and a serum half-life of 4.5 h, while phage particles displaying antibody fragments were reported to have serum half-life of less than 4 h [144,145]. Interestingly, Zou et al. reported a peptide phage display library to have a vastly shortened serum half-life of 20 min or less [100]. These studies emphasize the strong influence that the displayed peptide or antibody fragment has upon the pharmacokinetics of the nanoparticle. Phage extravasation is thought to be restricted by the continuous endothelium of capillaries, especially in tissues with tight cell–cell junctions. Examples of these are skeletal muscle, skin, connective tissues, and the brain [100,144]. In contrast, organs of the reticuloendothelial system have discontinuous endothelium, and contain open fenestrae as well as discontinuous basement membranes. Thus, the distribution and absorbance of phage particles into tissues are thought to be mostly regulated by the ultrastructure of the vasculature in different tissues. Given the heterogeneous and leaky nature of the tumor vasculature [144,146], it is possible that by 24 h post-injection, the phage may diffuse into the tumor interstitium, although high tumor interstitial pressure may reduce such penetration [144]. The enhanced permeability and retention (EPR) effect allows for an increased accumulation of particles at tumor sites and can be further enhanced by combining it with ligand-mediated targeting. The display of peptides on the phage can thus also improve their pharmacokinetic behavior, increasing their chances of interaction and accumulation at the target site.

#### 4.2.3. Clearance

Many studies have determined that the main route of phage clearance is via the liver [144]. Specifically, the hepatic uptake is attributed to Kupffer cells. Immunohistological data reveal phage coat proteins in the feces by 24 h, but the phage titer usually remains low at this timepoint. This suggests that the feces probably contain fragments of phage proteins, presumably excreted via the bile duct. Interestingly, most biodistribution reports reveal closely correlated phage infectivity and immunohistochemistry staining in most of the tissues investigated. The only exception is the poor correlation between staining and titer data derived from the feces, suggesting that the main site for phage metabolism and breakdown is in the liver. Some evidence exists of the phage being cleared through the kidneys. Yip et al. showed glomeruli immunohistology staining for phage coat proteins at timepoints as early as 5 min, followed by positive staining in the proximal tubule by 24 h [144]. 

#### 4.2.4. Effect of Size, Shape, and Charge upon Phage Pharmacokinetics

Nanomedicine is an advancing field that has led to breakthrough innovations in various drug delivery methods. Nanoparticles have increasingly become the focus of research as they offer unprecedented control over particle size, shape, and surface properties for downstream applications; in particular, filamentous phage particles have attracted significant attention as potential vehicles for drug delivery due to their unique morphology and desirable mechanical properties. The filamentous fd/M13 virus particle is ~6.6 nm in diameter and up to ~900 nm in length [147]. This elongated cylindrical virion has ~2700 coat protein VIII with a surface charge density of 0.46 eq/nm^2^ at pH 7.4 [147,148]. This leads most researchers to hypothesize that the major contributor to intrinsic phage effects is that of electrostatic interactions. But this characteristic is also what lends the phage particle resistance to a range of pH (2 to 11), chaotropic agents, high salt concentrations, and other harsh conditions. This structure of the phage also creates a flexible but strong particle. The measure of a persistence length is a basic mechanical property that quantifies the stiffness or rigidity of the polymeric capsid [149]. Interestingly, a filamentous bacteriophage is considered more rigid than single DNA strands but softer than microtubules [150]. The persistence length of a phage virion is 1265.7 ± 220.4 nm and has an elastic stretching modulus of ~50 MPa [150]. This is important as the dynamics of the rods in laminar flows are strongly affected by their flexibility. Thus, it is postulated by Driessen et al. that the elongated phage virion would display a pole-vaulting motion along the vascular wall within the hydrodynamic forces of physiological laminar flow [150,151]. This type of motion would predispose an interaction between the tip of the virion displaying coat protein III with the endothelial cells of the vascular wall [151]. As a result, the slender filamentous phage shape favors binding regardless of its orientation, as long as the tip of the virion is exposed to the endothelium. Further, when considering the drag force, wall shear rate, and the loading rate for a rod-shaped virion the rupture forces needed are such that a phage virion seems to be able to remain bound to its targeted biomarker under most conditions [152,153]. In fact, because of the flexibility of the virion, it is assumed that the bound phage would tend to rotate and be pushed down to the cell membrane without detaching from the target. This, in turn, would allow the phage to be quickly and easily internalized. Several studies have shown that phage-displayed peptides can be internalized by the target cells in a receptor-mediated manner [154]. However, the factors that influence the binding and internalization of phage-displayed peptides are complex and not well understood. Driessen et al. have developed a mathematical model that predicts the binding and receptor-mediated internalization of phage-displayed peptides based on mass balance equations that describe the kinetics of phage binding to cell surface receptors and subsequent internalization [150,155]. The binding process is characterized by both forward and reverse reaction rates, while internalization is characterized by a single forward rate. The model predicts that the ratio between the concentration of bound phage and initially injected phage is proportional to the number of receptors available and inversely proportional to K_d_ (dissociation constant). 

The role of intermolecular forces in biological processes cannot be overstated. Electrostatic, Van der Waals, and hydrophobic forces are all crucial contributors to the total free energy of these processes [156,157]. Understanding these forces is key to making informed decisions regarding targeting and delivery of therapeutic agents, as well as optimizing the pharmacokinetics of potential treatments. In particular, diffusion and extravasation can be achieved through active or passive targeting of organs or diseased vasculature [158]. Researchers must carefully consider the intermolecular forces involved in the process. These forces will play a key role in determining how the phage interacts with the target tissue, as well as how it is distributed throughout the body. For example, hydrophobic interactions may play a significant role in the binding of the phage to the target tissue, while Van der Waals forces may play a role in the distribution of the phage throughout the body [150,159]. In addition to understanding the intermolecular forces involved in the process, researchers must also carefully consider the pharmacokinetics of the phage itself [159]. This includes factors such as the rate of clearance from the bloodstream, the rate of internalization by the target cells, and the overall bioavailability of the agent. 

Further, multivalent ligand–receptor interactions are a fundamental aspect of biology, characterized by the binding of multiple ligands to multiple receptors. These interactions are seen in a variety of natural processes, such as virus attachment, where the virus utilizes multiple ligands to attach to host cells that have several receptors [159]. The use of multivalent interactions in biotechnology has been of great interest, and phage display is a method used to achieve this [122]. The use of multivalent interactions in phage display can lead to higher binding affinities than monovalent interactions by stabilizing the conformation, statistical rebinding, and clustering of receptors [160]. In addition, stochastic modeling studies have shown that a multivalent phage can survive more easily during selection than a monovalent phage, contributing greatly to the successful use of phage display in vivo [161]. It is hypothesized that multivalency plays an important role in recognizing multiprotein complexes and altering the proximity between receptors. The use of multivalent interactions in phage display has many applications in biotechnology, including drug discovery, diagnostics, and vaccine development. Furthermore, the high binding affinities and selectivity of multivalent interactions make them a promising approach for the identification of disease markers and the production of therapeutic agents.

#### 4.2.5. Pharmacodynamics of Phage Display Selected Peptides 

To date, most in vivo phage display selections and other subsequent in vivo experiments have focused mainly upon the pharmacokinetics of the phage particle itself and how the targeting ligand can modify the biodistribution of the particle. For example, BCP-1 and BCP-2 peptides prolong the blood circulation time, and RCC1-02 increases kidney clearance of phage (Table 3) [78,84]. Table 3 contains many examples of phage display-selected peptides with pharmacodynamic effects, from internalization by various organs to translocation across the blood–brain barrier or the intestines, all with tissue or cell type-specific homing abilities. Another example of a phage display-selected peptide with pharmacokinetic effects is a new class of peptides that can form transient pores or penetrate the skin [162,163]. The majority of phage clones identified with pharmacodynamic capabilities are identified via phage display selection in situ against cells, or in vivo against whole tissues. Hypothetically, the advantage of in situ and in vivo selections is the different possible conformations of a cell surface target due to multiple factors, including the presence of binding partners, and protein complexes, as well as variations in post-translational modifications. 

## 5. Conclusions

Phage display technology continues to be a versatile and widely utilized method for discovering peptides that target disease-related biomarkers via biopanning. While biopanning is efficient, it requires careful optimization of the various parameters to obtain specific and high-affinity peptides. Experimental factors, such as the type of the phage display library, phage and antigen concentrations, and incubation and elution conditions must be considered. Of the utmost importance is the choice of utilizing in vitro, in situ, or in vivo affinity selections, which is known to strongly influence the stability and selectivity, as well as the biodistribution and clearance of both peptides and peptide-conjugated nanoparticles. Current efforts by researchers to maximize pharmacodynamic effects through both active and passive targeting strategies will continue to drive this field of research. Overall, understanding and utilizing these mechanisms is crucial for developing effective targeted drug delivery systems. By thoughtfully considering these factors, researchers can develop peptides with enhanced affinity, specificity, and the desired pharmacokinetic profiles.

## Figures and Tables

**Figure 1 viruses-16-00570-f001:**
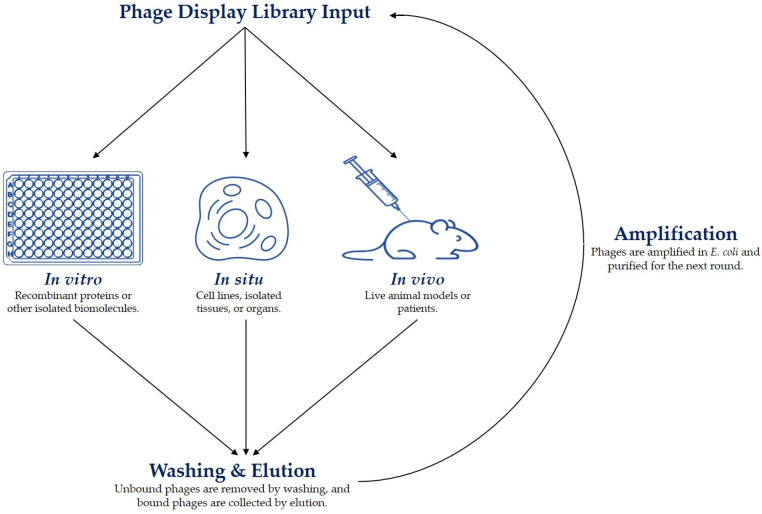
Phage display selection strategies.

**Table 1 viruses-16-00570-t001:** Comparison of phage display vectors.

Vector Type	Coat Protein	Displayed Molecules (n)	Comments
3	pIII	5	The type 3 vectors carry only one modified pIII gene.
33	pIII	1–3	The type 33 vector systems carry both a wild-type pIII phage gene and a modified pIII gene.
3 + 3	pIII	~1	The vector systems referred to as 3 + 3 have a modified pIII gene on a phagemid and utilize helper phage to introduce an additional wild-type pIII gene.
8	pVIII	2700	The type 8 vectors carry only one modified pVIII gene.
88	pVIII	~100–200	The type 88 vector systems carry both the wild-type pVIII phage gene and a modified pVIII gene.
8 + 8	pVIII	~100–200	The 8 + 8 vector systems have a modified pVIII gene on a phagemid and utilize helper phage to introduce an additional wild-type pVIII gene.

**Table 2 viruses-16-00570-t002:** Comparison of phage display biopanning strategies.

Strategy	Targets	Advantages	Disadvantages
In vitro	Recombinant proteins and other isolated biomolecules.	Easy setup, fast, cheap, low K_d_.	Poor pharmacokinetics, low specificity, low stability.
In situ	Cell lines, organoids, ex vivo tissues and organs.	Reflects the complex milieu of a cell, low-to-mid range K_d_, enhanced binding specificity and stability.	Higher K_d_ compared to in vitro selections.
In vivo	Live animals and patients.	Reflects the complexity of an in vivo milieu, enhanced pharmacokinetics, binding specificity, and stability.	Mid-range K_d_, expensive, time-consuming.

**Table 3 viruses-16-00570-t003:** Peptides from phage display.

Tissue/Organ	Name	Target	Effect	Sequence	Reference
Cancer-Immune Interactions	PD-L1 Pep-1 PD-L1 Pep-2	Programed Cell Death Ligand 1 (PD-L1)	Tumor Homing/ Blocking of T cell Function	CLQKTPKQC CVRARTR	[164]
Peripheral Blood Cells	BCP-1 BCP-2	RGD Integrin Targeting on Peripheral Blood Cells	Prolong Circulation within Blood	CNARGDMHC CIVRGDNVC	[84]
Blood-Brain Barrier	Clone 7	Unknown	Nose to Brain Translocation Capability/ Olfactory and Brain Homing	ACTTPHAWLCG	[165]
Heart	Cardiac Targeting Peptide (CTP)	Unknown	Heart/Cardio Myoblast Homing and Internalization	APWHLSSQYSRT	[166]
Ischemic Heart	None	Unknown	Myocardium Damaged by Ischemia- Reperfusion	CSTSMLKAC	[167]
Lung	GFE-1	Membrane Dipeptidase (MDP)	Mouse Lung Vasculature	CGFECVRQCPERC	[168]
Lung Epithelial	LTP-1	Unknown	Pulmonary Epithelial Translocation	CTSGTHPRC	[169]
Lung Cancer	Pep1	Unknown	Lung Adenocarcinoma with Increased Uptake Post Radiation Treatment	CAKATCPAC	[170]
Non-Small Cell Lung Cancer	Thx	Unknown	Non-Small Cell Lung Cancer	ARRPKLD	[171]
Kidney/Kidney Cancer	None	Kidney Injury Molecule (KIM-1)	Kidney Cancer	CNWMINKEC	[172]
Kidney	RCC1-02	Unknown	Redistribution Towards Kidney Clearance/ Avoidance of Protein Reabsorption	AGGLSFGTRRFEIGR	[78]
Intestine	4–1 4–11	Unknown	Internalization by Normal Intestine	SGHQLLLNKMP SFKPSGLPAQSL	[173]
Intestine		Sequence Homology (HIV gp120)	Translocation Across the Intestine	YPRLLTP	[174]
Injured Intestine	4–5	Unknown	Internalized by Injured Intestine	ILANDLTAPGPR	[173]
Colon Cancer	CP15	Unknown	Colon Cancer	VHLGYAT	[175]
Peritoneal Metastasis of Gastric Cancer	pIII	Unknown	Homing to-and Prevention of- Metastases of Gastric Cancer	SMSIASPYIALE	[176]
Gastric Cancer	Peptide 1131	Kita-Kyushu Lung Cancer Antigen-1 (KK-LC-1)	Gastric Cancer	Not given	[177]
Liver Cancer	HCBP1	Unknown	Liver Cancer	FQHPSFI	[178]
Pancreas	None	Putative EphA4 Receptor/ Sequence Similarity with Ephrin-A Ligands	Pancreas Islet Cells	CHVLWSTRC CVSNPRWKC	[15]
Pancreatic cancer	RGR RSR KAA KAR VGV EYQ	Sequence Similarity with Various Proteins	Angiogenic Vasculature of Pancreatic Islet Tumorigenesis	CRGRRST CRSRKG CKAAKNK FRVGVADV CEYQLDVE	[179]
Pancreas	None	Unknown	Pancreatic Beta Cells	LNTPLKS	[180]
Bone Cancer	NF-1	Unknown	Osteosarcoma Vasculature	CTKPDKGYC	[181]
Bone Cancer	OSP-1	Putative Heparan Sulfate Proteoglycan	Osteosarcoma	ASGALSPSRLDT	[182]
Bone Marrow	None	Putative Sequence Similarity with CD84	Bone Marrow	STFTKSP	[183]
Adipose	TDA1	Unknown	Transdermal Targeting of Visceral Adipose Tissue	CGLHPAPQC	[184]
Skin/psoriatic lesions	Pep3D	Interferon- Alpha Receptor	Reduces Psoriasis Symptoms	CIGNSNTLC	[185]
Skin	T2 Peptide	Lipids	Skin Penetrating	LVGVFH	[92]
Skin	None	Unknown	Transient Pore Formation Within the Skin for Transdermal Protein Delivery	ACSSSPSKHCG	[163]
Skin	Skin Penetrating and Cell Entering (SPACE) Peptide	Unknown	Keratinocytes, Fibroblasts, and Endothelial Cells	ACTGSTQHQCG	[162]

## Data Availability

No new data were created.

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
