# Peer review of "Improving Pharmacokinetics of Peptides Using Phage Display"

_viruses, 2024, doi:10.3390/v16040570_

Round 1

Reviewer 1 Report

Comments and Suggestions for Authors

This paper is a quite a nice review on phage display, focussed on pharamacokinetics of peptides exposed on the phage virion surface. Although many reviews have been published recently in the field of phage display, pharamacokinetics of displayed peptides was not comprehensively discussed. The manuscript text is well-prepared and I recommend its accpetance after suitable revisions. Specific suggestions for revisions are indicated below.

1. Introducion, first paragraph: quite old literature is cited in this introductory paragraph. While it is reasonable to cite the work by Smith (1985), other papers are quite old. Since several review papers on phage display were published recently, it is recommended to cite some of them, either instead of or apart from those cited in the current version of the paper.

2. Presentation of some figures to illustrate the text would increase clarity of the presentation and make the paper more attractive to a general reader.

3. Line 82 - It is not necessary to provie abbreviated name of any species after presenting the full name. The commonly accepted rule is that the full name (like Escherichia coli) is written when preseting for the first time in the text, and then, abbreviated name (like E. coli) is used throughout the further text.

4. Chapter 3.1. - Please, describe shortly the 3, 3+3, 33, 8, 8+8, and 88 systems before discussing their use or cite appropriate recent literature. Otherwise, the text might be hard to follow, especially for those who are not experts in the phase display technology.

5. Line 573 - replace 35 S-methionine and 35 S-cysteine with 35S-methionine and 35S-cysteine, respectively.

Author Response

Thank you for the thorough and favorable review of our article. We have made the requested revisions according to your comments as shown below.  

1. The following references have been added to the introductory paragraph to include more recent publications:  

Islam MS, Fan J, Pan F. The power of phages: revolutionizing cancer treatment. Front Oncol. 2023 Nov 15;13:1290296. doi: 10.3389/fonc.2023.1290296. PMID: 38033486; PMCID: PMC10684691.  

Song BPC, Ch'ng ACW, Lim TS. Review of phage display: A jack-of-all-trades and master of most biomolecule display. Int J Biol Macromol. 2024 Jan;256(Pt 2):128455. doi: 10.1016/j.ijbiomac.2023.128455. Epub 2023 Nov 25. PMID: 38013083.  

França RKA, Studart IC, Bezerra MRL, Pontes LQ, Barbosa AMA, Brigido MM, Furtado GP, Maranhão AQ. Progress on Phage Display Technology: Tailoring Antibodies for Cancer Immunotherapy. Viruses. 2023 Sep 9;15(9):1903. doi: 10.3390/v15091903. PMID: 37766309; PMCID: PMC10536222.  

Pierzynowska K, Morcinek-OrÅ‚owska J, Gaffke L, Jaroszewicz W, Skowron PM, WÄ™grzyn G. Applications of the phage display technology in molecular biology, biotechnology and medicine. Crit Rev Microbiol. 2023 Jun 4:1-41. doi: 10.1080/1040841X.2023.2219741. Epub ahead of print. PMID: 37270791.  

Arab A, Nicastro J, Slavcev R, Razazan A, Barati N, Nikpoor AR, Brojeni AAM, Mosaffa F, Badiee A, Jaafari MR, Behravan J. Lambda phage nanoparticles displaying HER2-derived E75 peptide induce effective E75-CD8+ T response. Immunol Res. 2018 Feb;66(1):200-206. doi: 10.1007/s12026-017-8969-0. PMID: 29143917.  

Voulgaridou GP, Theologidis V, Xanthis V, Papagiannaki E, Tsochantaridis I, Fadouloglou VE, Pappa A. Identification of a peptide ligand for human ALDH3A1 through peptide phage display: Prediction and characterization of protein interaction sites and inhibition of ALDH3A1 enzymatic activity. Front Mol Biosci. 2023 Mar 20;10:1161111. doi: 10.3389/fmolb.2023.1161111. PMID: 37021113; PMCID: PMC10067601.  

Zhang L, Zhang S, Wu J, Wang Y, Wu Y, Sun X, Wang X, Shen J, Xie L, Zhang Y, Zhang H, Hu K, Wang F, Wang R, Zhang MR. Linear Peptide-Based PET Tracers for Imaging PD-L1 in Tumors. Mol Pharm. 2023 Aug 7;20(8):4256-4267. doi: 10.1021/acs.molpharmaceut.3c00382. Epub 2023 Jun 27. PMID: 37368947.  

2. Two tables (Table 1 and Table 2) and one figure (Figure 1) have been added.

3. The abbreviated name E. coli was deleted.  

4. A table (Table 1) was added to describe the 3, 3+3, 33, 8, 8+8, and 88 phage display systems.  

5. 35 S-methionine and 35 S-cysteine have been changed to 35S-methionine and 35S-cysteine, respectively.

Reviewer 2 Report

Comments and Suggestions for Authors

In this review, Asar et al. discussed how phage display technology can be used to improve the pharmacokinetics of peptides and peptide-conjugated nanoparticles for targeting disease-related biomarkers. The authors comprehensively review the principles and applications of phage display, the factors that influence the biodistribution and clearance of peptides, and the advantages and challenges of using phage display for peptide discovery and maturation, as well as the principles of pharmacokinetics for peptides. Here are some comments on this study:

1.        Abstract: The abstract should provide more information on the main challenges and limitations in characterizing the pharmacokinetics of phage-displayed peptides and specify the main contributions and novelties of this review.

2.        Introduction: A similar issue to the abstract is that the introduction should highlight the significance of this review by providing more background information on gaps or limitations in pharmacokinetic studies of phage display peptides.

3.        Line 81 “Ff filamentous” should be “F-specific filamentous”.

4. Sections 2 and 3, it is recommended that the author add some figures to illustrate this.

5.        Section 3.2, for different plasmids or strategies, could authors compare the differences and provide some suggestions?

6.        Section 3.4.1, it is suggested that the authors add a table to compare the advantages and disadvantages of the three methods.

7.        For a review paper, in addition to the collection of literature, it is more important to provide the author's point of view, evaluation, or recommendations for the field of study. It is proposed that authors provide some summaries and present their views and opinions on the phage display and pharmacokinetic for peptides.

8.        Reference 24, incorrectly formatted reference title.

Author Response

Thank you for the thorough and favorable review of our article. We have made the requested revisions according to your comments as shown below.  

  1. The abstract has been updated to provide more information on the main challenges and limitations in characterizing the pharmacokinetics of phage-displayed peptides.
  2. Changes have been made to the introduction to highlight the significance of the review by providing more background information on gaps or limitations in pharmacokinetic studies of phage display peptides. The following sections have been added/changed: Lines 48-53 and lines 82-86.
  3. Ff has been changed to Ff-specific.
  4. Table 1 and Figure 1 have been added.
  5. We have added information to provide some strategies to select the best vectors for phage display. Lines 199-202 and 212-213 have been added. 
  6. Table 2 which compares in vitro, in situ, and in vivo biopanning strategies has been added to section 3.4.
  7. We have added information to reflect our point of view to the following lines: 94-96, 302-317, 356-360, 411-418, 449-456, and 527-532.
  8. Reference 24 (now 27) has been corrected.